# Estimation and Closed-Loop Control of *COG/ZMP* in Biped Devices Blending *CoP* Measures and Kinematic Information

**Giuseppe Menga** [1,*,†] **and Marco Ghirardi** [2,†]

1   Department of Control and Computer Engineering, Politecnico di Torino, Corso Duca degli Abruzzi 24, 10129 Torino, Italy
2   Department of Management and Production Engineering, Politecnico di Torino, Corso Duca degli Abruzzi 24, 10129 Torino, Italy; marco.ghirardi@polito.it
*   Correspondence: giuseppe.menga@formerfaculty.polito.it; Tel.: +39-011-0907261
†   These authors contributed equally to this work.

**Abstract:** The zero moment point (*ZMP*) and the linearized inverted pendulum model linking the *ZMP* to the center of gravity (*COG*) have an important role in the control of the postural equilibrium (balance) of biped robots and lower-limb exoskeletons. A solution for balance real time control, closing the loop from the joint actual values of the *COG* and *ZMP*, has been proposed by Choi. However, this approach cannot be practically implemented: While the *ZMP* actual value is available from the center of pressure (*CoP*) measured under the feet soles, the *COG* is not measurable, but it can only be indirectly assessed from the joint-angle measures, the knowledge of the kinematics, and the usually poorly known weight distribution of the links of the chain. Finally, the possible presence of unknown external disturbance forces and the nonlinear, complex nature of the kinematics perturb the simple relationship between the *ZMP* and *COG* in the linearized model. The aim of this paper is to offer, starting from Choi's model, a practical implementation of closed-loop balance control fusing *CoP* and joint-angle measures, eliminating possible inconsistencies. In order to achieve this result, we introduce a model of the linearized inverted pendulum for an extended estimation, not only of *COG* and *ZMP*, but also of external disturbances. This model is then used, instead of Choi's equations, for estimation and balance control, using $\mathcal{H}_\infty$ theory. As the *COG* information is recovered from the joint-angle measures, the identification of a statistically equivalent serial chain (*SESC*) linking the *COG* to the joint angles is also discussed.

**Keywords:** biped robotics; exoskeletons; postural equilibrium; zero moment point; inverted pendulum; robust control

## 1. Introduction

The zero moment point (*ZMP*) and linearized inverted pendulum have continued to play a fundamental role in the control of postural equilibrium of biped robots and lower-limb exoskeletons since their introduction by Vukobratovic [1]. Vukobratovic showed that the center of pressure (*CoP*) of reaction forces under the feet soles on a flat horizontal surface coincides with a point he called the zero moment point (*ZMP*) and that postural equilibrium can be guaranteed if the *CoP* (alias *ZMP*) is maintained inside the convex hull of the surface encompassing the supporting foot (or feet in double stance). Moreover, a very simple relationship, based on the linearized inverted pendulum, links the *ZMP* and the center of gravity (*COG*) projection on the ground of the mechanical chain. The goal is to control the *COG* acting on the joint angles as both are algebraically linked by the kinematics of the chain, the target objective being the *ZMP*.

If motion maintaining postural equilibrium is desired (e.g., during a step in gait), to control the *ZMP* requires a certain degree of anticipation. Hence, in the so-called preview control framework [2], transition of *COG*, *COG* velocity, and *ZMP* were precomputed in advance and applied in open loop as reference to the biped control. However, this approach has not been able to model a closed-loop system and does not face the problems of disturbance rejection and stability.

A solution to track the preview trajectories in closed loop was successively proposed by Choi [3]. The preview *COG* − *ZMP* trajectories during rectilinear gait were reviewed, and a closed-loop strategy was devised and proven, using Lyapunov techniques, to guarantee closed-loop stability and a bounded error tracking of the *COG* and *ZMP* preview references. The loop was closed from the actual values of *COG* and *ZMP*, generating a feedback signal to control the *COG* velocity. Then, a speed control for the joints from the *COG* velocity was designed using a *COG* Jacobian with specified embedded motion. The output measures were not detailed, but presumably, *COG* was assessed by measuring the joint angles from the weight distribution on the kinematic chain, and *ZMP* was measured by pressure sensors under the feet of the biped. The strategy, according to theory, guarantees closed-loop stability. However, when the authors tested it in simulations and practical examples, it showed a lack of robustness to disturbances or poor damping of the closed-loop dynamics. Lyapunov theory guarantees stability but does not say how much the resulting closed-loop poles will be damped.

If real-time measures (let us call them $COG_m$ and $ZMP_m$) of *COG* and *ZMP* are independently available, it is reasonable to assume that before closing the loop, a filtering is performed, fusing both data. However, if these are generated, as stated before, the latter by direct measures and the former indirectly from joint measures and a priori information, they are not always consistent with the relationship stated by the linearized inverted pendulum. The main reasons are: uncertainties in the model parameters (especially in the weight distribution when dealing with an exoskeleton interacting with a patient), external forces acting on the biped (crutches or a chair in a sit-to-stand exercise), centrifugal forces in the frontal plane when motion is not rectilinear.

This study was motivated by the intention to improve postural equilibrium in lower-limb exoskeletons for rehabilitation. The same approach is at the basis of all applications needing balance control of biped robots, such as biped walking in rectilinear [4,5] and curved trajectories [6], in haptic lower-limb exoskeletons [7], and in performing sit-to-stand exercises [8], described by the authors in other papers.

## 1.1. Paper Contributions

The main contribution of the paper is the development of a feedback control more robust than the one offered by Choi. In order to achieve this result, a detailed understanding of the closed-loop dynamics generated by controlling an inverted pendulum is presented, with particular attention devoted to the control design techniques and the engineering problems in closing the loop. Then, in order to make the filtering effective and close the loop from $COG_m$ and $ZMP_m$, ensuring compatibility, the proposed approach operates at two levels: a nonlinear algebraic function and a a linear dynamic model.

The nonlinear function is a simplified mapping from joint angles to *COG*, called statistically equivalent serial chain (SESC) [9], to be identified in a priori experiments. As this identification is based on the same force sensors under the feet used for measuring the *CoP*, it also resolves any calibration mismatch.

The linear model is an extended system based on the inverted pendulum from input *u*, the reference *COG* velocity, and output $COG_m$ and $ZMP_m$, used to estimate, along with the *COG*, *ZMP*, and external disturbances affecting the *CoP*, the model states. Then, using the estimated states, the loop from $COG_m$ and $ZMP_m$ can be closed applying robust control theory. The reasons for estimating the unknown external force disturbances are twofold: (1) to take into account real external disturbances, especially in exoskeletons but also in biped robots (e.g., centrifugal forces in turning

while walking); (2) robustness in the $COG - ZMP$ joint estimation, accommodating modeling errors, parameter uncertainties, and the simplifications introduced by the linearized inverted pendulum.

Still recently, the linearized inverted pendulum has continued to be at the basis of the models for balance control ([10,11] and references therein). However, to the authors' knowledge, there are no works introducing, for robustness, an extended system to estimate disturbances, or the need for a SESC identification.

The proposed control is a non-conventional tracking problem, as two separate model states are tracked. Two different control design techniques are proposed and tested to control the extended system: (1) computing a robust estimator and solving the output feedback problem from the estimated states using a numerical approach based on the Levenberg–Marquardt algorithm [12,13]; (2) solving the standard robust regulator, adapted to deal with the preview signal tracking.

In order to test the approaches, three different experiments were performed. First, both observer and state feedbacks were implemented and compared with Choi's original feedback through simulation of the 2D linearized inverted pendulum. In a standing position, a preview shift of the $COG$ from the heels to the tips of the feet was imposed, while in the meantime, an external force disturbance was applied. Then, a real lower-size mechanical mock-up was considered, composed of foot, leg, thigh, and trunk, with three degrees of freedom (DOF) in the sagittal plane to represent the real exoskeleton for implementing the sit-to-stand exercise. The SESC model of this simplified kinematics was identified with a priori experiments and used in the proposed feedback control through the $COG$ Jacobian of the chain. Finally, a non-linear simulation of the full-scale exoskeleton with patient was run on the same exercise executed by the linearized inverted pendulum, emulating the first phase of a stand-to-sit exercise. A complete sit-to-stand exercise with the presence of a chair and switching dynamics exploiting the same control technique can be found in [8].

The paper is organized as follows. Section 2 reviews Choi's results. Section 3 introduces the main contribution of the paper: a $COG - ZMP$ model of the linearized inverted pendulum and an extended system, embedding in the model external disturbances, for applying robust estimation and robust control. This model is also used in the Appendix to show the limitations of Choi's feedback. Sections 3.1 and 3.2 present the robust estimator–estimate state feedback and the standard robust regulator. Sections 3.3 and 4 contain simulated and real control experiments. In particular, Section 4.1 approaches the identification of the SESC model, and Section 4.2 presents the simulation of a stand-to-sit exercise. Section 5 concludes the paper. The appendix discusses the limitations of Choi's original method.

## 2. Choi's Approach

As in [3], the 3D linearized inverted pendulum is split into two separate, independent 2D models for the sagittal and the frontal planes. However, in this paper only the sagittal plane will be considered, with axes $x$ (horizontal) and $z$ (vertical). The equation linking $COG$ and $ZMP$, adopting Choi's notation, is

$$p = c - (1/w_n^2)\ddot{c}, \tag{1}$$

$$w_n \triangleq \sqrt{g/c_z}, \tag{2}$$

where $p$ is the coordinate of the $ZMP$, $c$ is the projection of the $COG$ on the ground, $c_z$ is the height of the $COG$, and $g$ is the acceleration of gravity. $w_n$ is the only parameter of the model of the simplified biped walking system.

Let $p_d$, $c_d$, and $\dot{c}_d$ indicate the desired preview trajectories of the $ZMP$, of the $COG$, and of its derivative during a postural exercise, and assume that the pendulum joint servo is controlled in speed by an input signal $u$ according to

$$\dot{c} = u + \epsilon, \tag{3}$$

where $\epsilon$ accounts for the speed-tracking error and process disturbances and $u$ is given by the following feedback law:

$$e_c = c_d - c, e_p = p_d - p, u = \dot{c}_d + k_c e_c - k_p e_p. \qquad (4)$$

Then, Choi's results prove, with Lyapunov theory ([3], Theorem 1), that if $k_c > w_n$ and $0 < k_p < w_n$, the closed-loop system is bounded disturbance ($\epsilon$) - bounded errors ($e_c, e_p$) stable.

Anyway, in spite of stability, a feedback implemented using Choi's equations has a poor damping of the closed-loop dynamics in practical operating conditions. A proof and discussion about this topic can be found in Appendix A.

## 3. An Extended System for COG–ZMP Robust Estimation and Control

Choi does not introduce any input–output dynamic model to prove his results, but only a Lyapunov function directly based on Equations (1)–(4). Here, vice versa, the essence of the feedback control problem involving measures of *COG* and *ZMP* with the reference velocity as input is captured by the simple model of the block diagram of Figure 1: a third-order model with states $c, \dot{c}, \ddot{c}$, where the jerk of the *COG* (in the following with an excess of notation, *COG* indicates its projection on the ground) is controlled by a reference velocity signal $u$ in an internal partial speed loop with velocity gain $k_v$ and output *COG* and *ZMP*. The third-order model is needed to guarantee a realistic strictly proper system for the design of the state estimator and feedback as position and acceleration are both present in the output, and in the meantime, representing the internal speed loop with the servo dynamics. If the gain of the local speed loop is taken relatively high ($k_v > 100$) the *COG* speed will closely track the reference $u$, as desired. This model does not take into account external forces acting on the system or internal disturbances, as in the case of a lower-limb exoskeleton with the presence of a patient. External forces are introduced when crutches are used or when the patient is sitting on a chair in a sit-to-stand exercise, or simply to accommodate discrepancies between *COG* and *ZMP* measures. Internal disturbances are generated by the involuntary motion of the patient in the small freedom offered by the exoskeleton, independent of the joint motion, and obviously, by modeling errors.

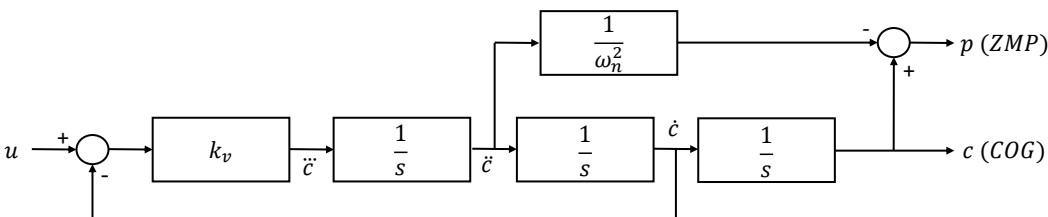

**Figure 1.** The COG–ZMP model of the linearized inverted pendulum.

Then the model can be completed, generating the extended system of Figure 2, as defined in robust control theory (for the definition of the extended system and its role in robust control and $\mathcal{H}_\infty$ theory, see [14,15]).

$F$ represents low-frequency external forces influencing the *CoP*. In the model $ZMP_{actual}$ (i.e., *CoP*), the value measured and $ZMP_{ideal}$, the one linked to the *COG* by the linearized inverted pendulum relationship, are defined separately, where $\delta$ is the difference between the two, the effect of disturbance $F$ to be estimated. *COG* and *CoP* are measured as before, taking into account measurement noise represented by two high pass filters $W_{noise_{COG}}, W_{noise_{ZMP}}$. Output objectives are set on the *COG* and on the *ZMP* for sensitivity requirements with respect to process noise $\epsilon$ (in a different context, here $\epsilon$ has the same interpretation and scope as in Equation (3)) and the effect of the unknown external force $F$. The weighting functions $W_{COG}$ and $W_{ZMP}$ are chosen to guarantee steady-state gain (i.e., tracking error with respect to disturbances $\epsilon$ and $F$) and the frequency band of the loop transfer function in the designed feedback. In order to have a balanced design, the control activity $z_u$ (with a weighting function $W_u$) is added as an objective against measurement noises $n_{COG}$ and $n_{ZMP}$, to set the control

activity. This extended system is used to design robust estimators of $COG$, $\dot{COG}$, $ZMP_{ideal}$, and $\delta$, as well as robust controls.

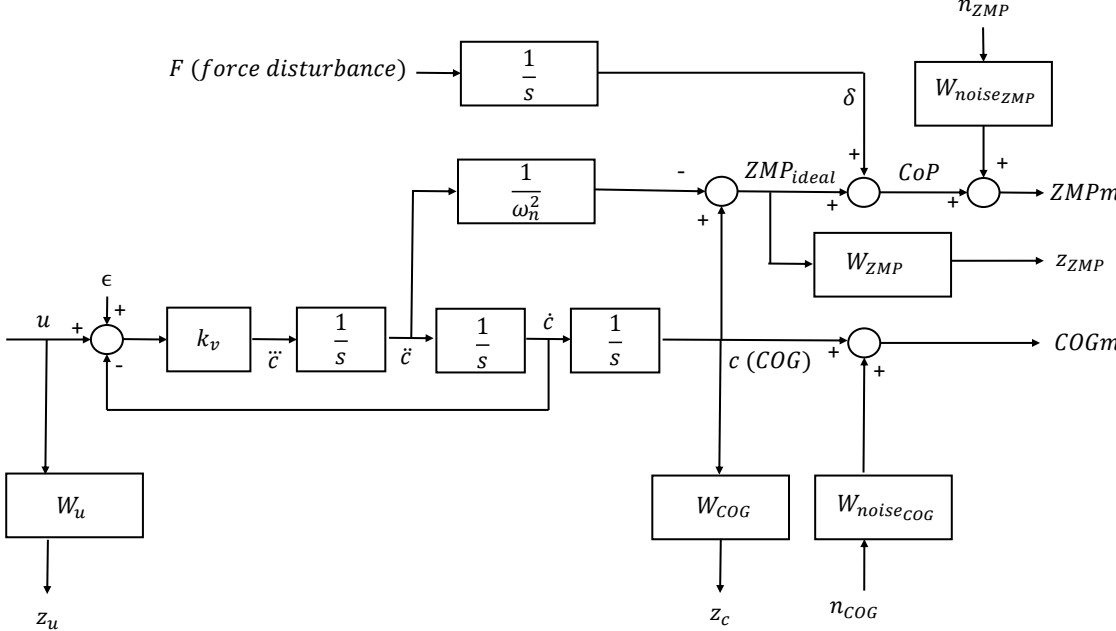

**Figure 2.** The extended system of COG–ZMP, disturbance model of the linearized inverted pendulum.

Let $\hat{c}, \dot{\hat{c}}, \hat{p}, \hat{\delta}$ be the estimates of the $COG$ projection on the ground, its derivative, the $ZMP_{ideal}$, and $\delta$. Then the control strategy of Equation (4) is modified as follows:

$$
\begin{aligned}
e_c &= c_d - \hat{c} - \hat{\delta}, \\
e_{cv} &= \dot{c}_d - \dot{\hat{c}}, \\
e_p &= p_d - \hat{p} - \hat{\delta}, \\
u &= \dot{c}_d + k_c e_c - k_p e_p + k_{cv} e_{cv},
\end{aligned}
\tag{5}
$$

with the control scheme represented in Figure 3. This means that $c$ (and in the steady state, $p$) must track a perturbed reference in order to guarantee that the $CoP$, and not the $ZMP_{ideal}$ linked to the $COG$, follows the desired preview signal, despite $ZMP_{ideal}$ and $COG$ converging to the same value in the steady state, independently of the presence of disturbances. A feedback from $\dot{\hat{c}}$ is also introduced as it has a critical influence on the closed loop damping.

In the next subsections, two different approaches, based on robust control theory, to compute the state observer and the state feedback coefficients, are introduced and tested. The estimates and the coefficients in Equation (5) result explicitly from standard $\mathcal{H}_\infty$ techniques by operating a state-space transformation in the extended system of Figure 2, choosing as states $c, \dot{c}, p, \delta$, augmented (for the whole extended system) by the unobservable or uncontrollable states introduced by the dynamics of the weighting functions.

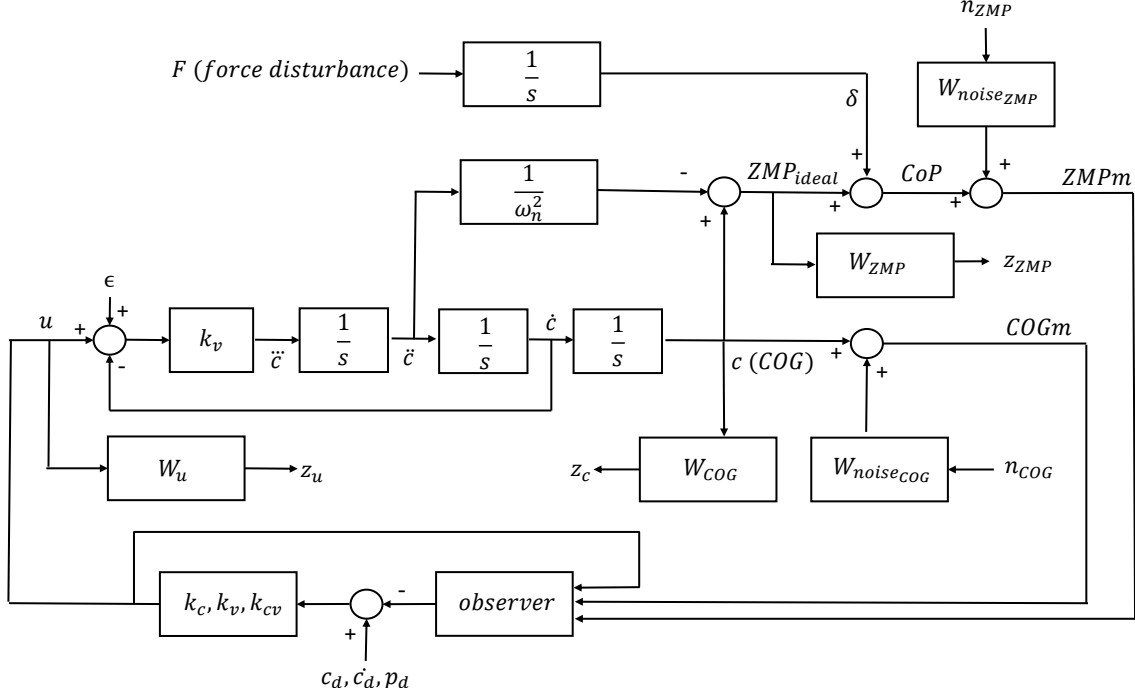

**Figure 3.** Control of COG–ZMP with observer feedback.

### 3.1. Separate Estimator and Feedback

In this first approach, the extended system with the new state representation is used to design a state estimator.

Then, a static output feedback problem for constrained pole placement is solved on the cascade of the extended system and observer to derive the gains $k_c$, $k_{cv}$, and $k_p$ in (5).

It is known that static output feedback has no analytical solution. Hence, a numerical technique based on the Levenberg–Marquardt algorithms was implemented. No algorithm details are presented here. Just note that by minimizing the sum of the squares of a certain number of penalty functions, the closed-loop poles are brought into a stability region with desired damping, constraining $k_c$ to be greater than a lower bound, and the control activity (measured by the $\mathcal{H}_\infty$ norm of the closed-loop operator from measuring noises to control objectives) to be smaller than an upper bound.

### 3.2. $\mathcal{H}_\infty$ Robust Control

In the second approach, estimation and feedback are jointly computed solving a $\mathcal{H}_\infty$ robust control regulator. Even if the classical separation of $\mathcal{H}_2$ does not apply in $\mathcal{H}_\infty$ controls [14,15], a weakly coupled state observer and estimated state feedback can still be recognized. Then, maintaining the feedback coefficients of the estimated states of interest ($\hat{c}, \dot{\hat{c}}, \hat{p}, \hat{\delta}$) and setting to zero the remaining ones relative to unobservable or uncontrollable modes (it can be verified that this has very little influence on the closed-loop poles), the strategy of Equation (5) can be implemented. It is interesting to note the similarity of the performances obtained between the two approaches, as shown in the next section.

### 3.3. Simulation Results

The model of an exoskeleton with a patient used in [5,7] was considered, with parameter $w_n = 3.34$ and choosing a speed gain for the velocity loop of $k_v = 250$. In this experiment, the linear model of the block diagram of Figure 2 was simulated. In an erect posture, with a preview reference computed as suggested by [2], a step transition on the sagittal plane of $p$ of 0.2 m moving the $c$ from the heels to the tips of the feet at time $t = 1.25$ s was imposed. Then at $t = 3.5$ s, an external horizontal force

disturbance acting in the sagittal plane on the *COG*, tries to move the *CoP* outside the feet support by an additional 0.1 m, causing—if not compensated—a loss of balance.

The experiments are compared using the identical tracking gain $k_c$ of the *COG* loop and the best (for damping) value of $k_p$ with Choi's control and feedback from a robust estimator obtained from the extended system of Figure 2. The gain parameters adopted in the case of Choi's feedback were $k_c = 60$, $k_p = 5$, with a resulting damping ratio of the dominant poles of 0.05; and in the case of observer feedback, $k_c = 60$, $k_p = 3$, $k_{cv} = 4$, with a resulting damping ratio of the dominant poles of 0.7.

The figures represent reference (dashed) and actual *COG* (blue), $ZMP_{ideal}$ (red), and *CoP* (cyan), *COG* speed estimate (green), and the estimate of the disturbance effect (violet).

Figure 4 shows the results adopting the Choi control. The low damping of the closed-loop poles is clearly visible. Note that, ignoring the disturbances, the *CoP* does not follow the reference path and exits from the tip of the feet.

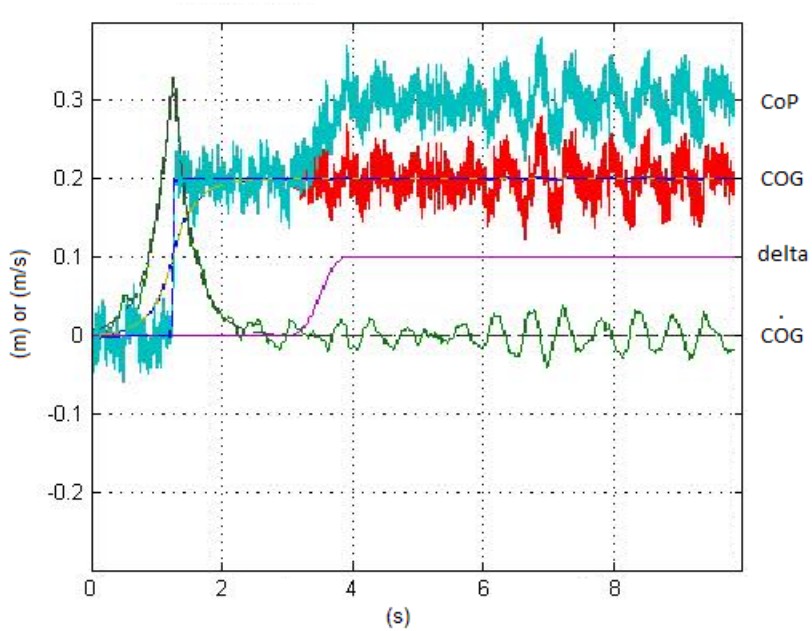

**Figure 4.** Choi's control without any disturbance compensation.

Vice versa, when disturbances are also estimated, after a short interval of time depending on the filtering bandwidth of the estimator, the *ZMP* returns to the desired value. Figures 5 and 6 show the feedback from the extended estimator with compensation of disturbances, obtained with the approaches of Sections 3.1 and 3.2, respectively.

The transition of the force disturbance was chosen to be unrealistically steep to evidence that, because of the estimator bandwidth, the compensation of the disturbance can't be perfect, depending on the values assigned to the weighting functions in the extended system.

The delay in the estimation of $\delta$, as a consequence of the disturbance, is shown in Figure 7, where $\delta$ is in blue and its estimate is in green.

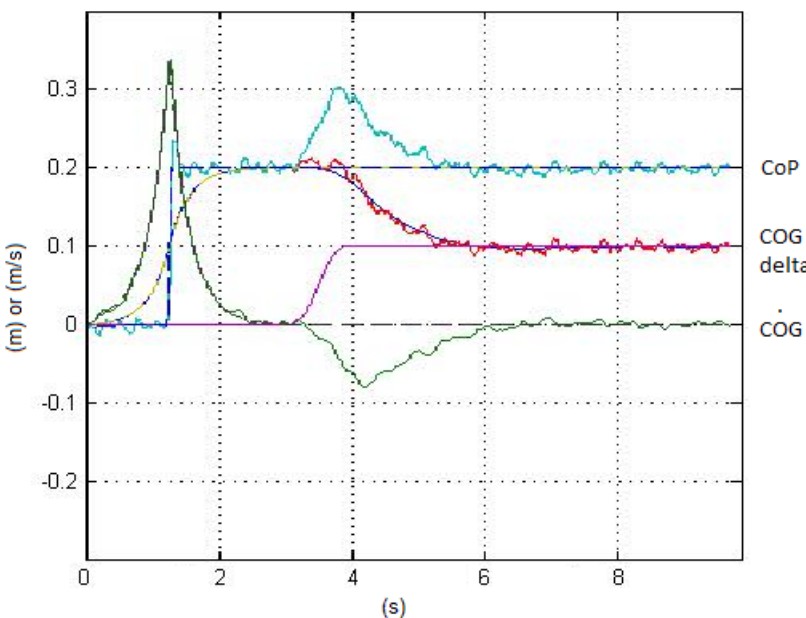

**Figure 5.** Extended observer feedback with disturbance compensation.

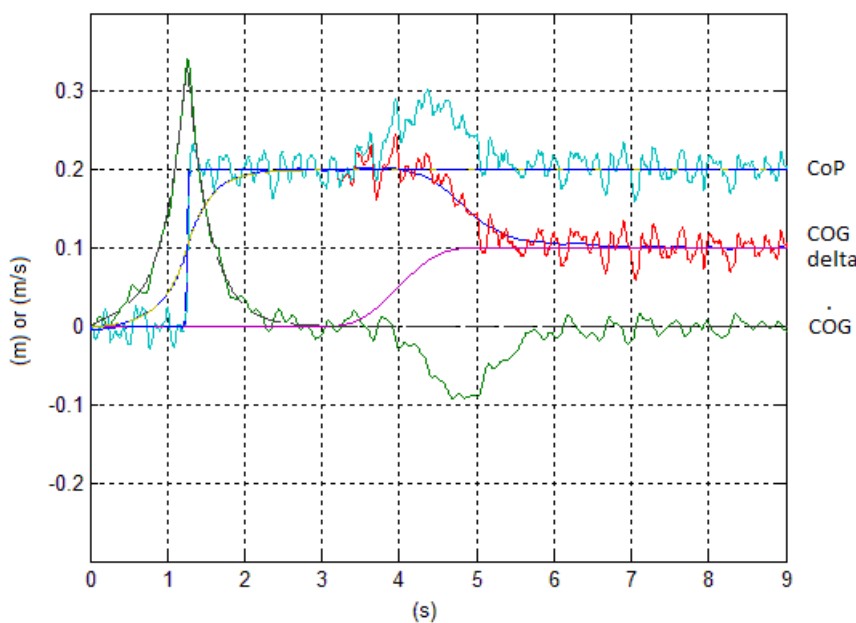

**Figure 6.** Robust control with disturbance compensation.

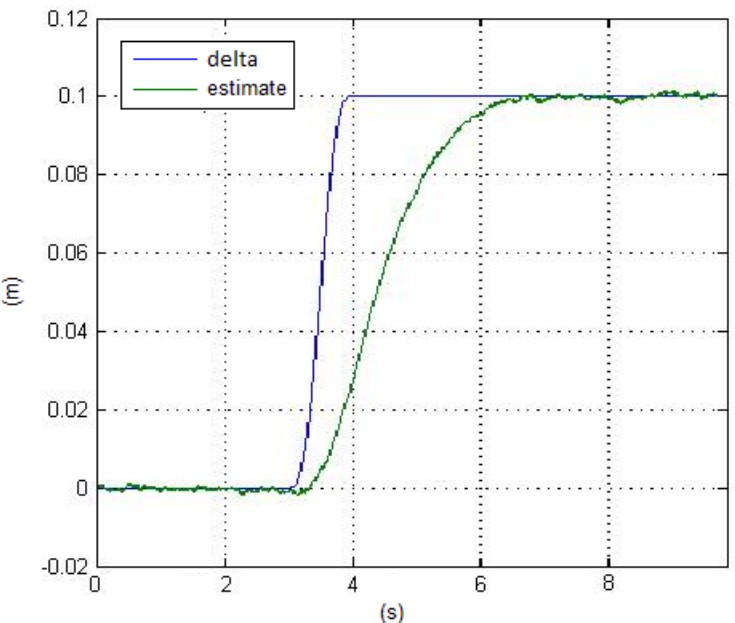

**Figure 7.** Effect of disturbance on the *CoP* and its estimation.

## 4. Control of a 3 DOF Biped

The approach was tested on the real three-DOF small-scale mock-up of an exoskeleton (Figure 8) and on the simulation of the full-scale exoskeleton. The chains, in both cases, are composed of two joint feet, legs, thighs, and one trunk. The $3 \times 3$ Jacobian matrix relating *COG* to joints embedding knee motion and trunk attitude is the following:

$$\begin{bmatrix} \dot{c} \\ \dot{\theta}_2 \\ \dot{\theta}_{trunk} \end{bmatrix} = \begin{bmatrix} & J_{cog} & \\ 0 & 1 & 0 \\ 1 & 1 & 1 \end{bmatrix} \cdot \begin{bmatrix} \dot{\theta}_1 \\ \dot{\theta}_2 \\ \dot{\theta}_3 \end{bmatrix}, \tag{6}$$

where $\theta_1, \theta_2, \theta_3$ are the angles of the ankle, knee, and hip, $J_{cog}$ is the Jacobian of $c$, and $\theta_{trunk}$ is the attitude of the trunk. Joints are controlled by velocity servos, with their references being obtained through the inverse of the Jacobian matrix (6) driven by speed feedback signals. The *COG* speed feedback is similar to the one used for the linearized inverted pendulum (5), where the measures of the $ZMP_m$ were obtained from the *CoP* and that of the $COM_m$ from the joint-angle measures $\theta_{1m}, \theta_{2m}, \theta_{3m}$. The remaining two feedbacks, from the knee angle and trunk attitude measures $\theta_{2m}, \theta_{trunk_m}$, are simply proportional feedbacks, the last measure being obtained from an inertial sensor:

$$\begin{bmatrix} u_{\theta_2} \\ u_{\theta_{trunk}} \end{bmatrix} = \begin{bmatrix} k_{knee} \cdot (\theta_{2_{ref}} - \theta_{2_m}) \\ k_{trunk} \cdot (\theta_{trunk_{ref}} - \theta_{trunk_m}) \end{bmatrix}, \tag{7}$$

where $\theta_{2_{ref}}$ and $\theta_{trunk_{ref}}$ are the references chosen according to the desired postural exercise.

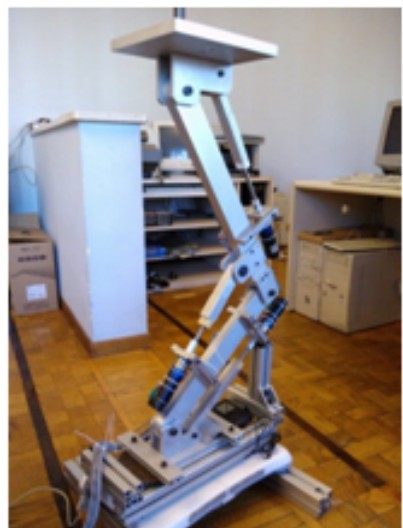

**Figure 8.** A small-scale 3 degrees of freedom (DOF) leg of the exoskeleton.

### 4.1. Joint Angles–$COG_m$ Map Fitting and Control

In order to apply the results of the previous section to a biped device, it is necessary to have a reliable evaluation of the *COG* from the joint angles, consistent with the *CoP*. Its value depends on the position of the center of masses and weights of each link of the chain. Those data are scarcely known in advance but can be identified with a series of a priori experiments. This approach is called statically equivalent serial chain (SESC) modeling [9] (see also [16,17] for applications to rehabilitation).

Espiau et al., in [9], showed from experiments measuring the projection of the *COG* on a force table that the physical parameters of a kinematic chain cannot be identified uniquely. What can be identified is only a set of expressions of them, representing classes of equivalent (with respect to the *COG*) chains. These expressions appear in linear form in the SESC model. Considering a three-joint kinematic model composed of feet, leg, thigh, and HAT (head, arms, trunk) for motions in the sagittal plane, the parameters of the SESC model can be identified using least squares with two equations and two sets of experiments: collecting and recording a series of joint-angle positions with the corresponding measures of the *CoP* in the steady state and a set of samples of joint angles and *CoP* trajectories in motion spanning the operating area at random. A slight modification of the model presented in [16] is proposed here, where the first equation, expressing explicitly $COG_x$, refers to steady-state experiments, while the second equation, expressing $COG_z$ indirectly (Equations (1) and (2) can be rewritten as $COG_x - ZMP_x = C\ddot{O}G_x/9.81 \cdot COG_z$), refers to dynamical ones.

The equations are:

$$\begin{bmatrix} COG_x \\ COG_x - ZMP_x \end{bmatrix} =$$

$$\begin{bmatrix} 1 & 0 \\ 0 & C\ddot{O}G_x/9.81 \end{bmatrix} \cdot$$

$$\begin{bmatrix} 1 & 0 & sin(\theta_1) & sin(\theta_1 + \theta_2) & sin(\theta_1 + \theta_2 + \theta_3) \\ 0 & 1 & cos(\theta_1) \cdot b & cos(\theta_1 + \theta_2) \cdot b & cos(\theta_1 + \theta_2 + \theta_3) \cdot b \end{bmatrix} \cdot$$

$$\begin{bmatrix} r_{1x} \\ r_{1z} \\ r_2 \\ r_3 \\ r_4 \end{bmatrix}, \tag{8}$$

with

$$r_{1x} = (m_0 \cdot x_0 + (m_1 + m_2 + m_3) \cdot x_1)/m_{tot},$$
$$r_{1z} = (m_1 + m_2 + m_3) \cdot z_1/(m_1 + m_2 + m_3),$$
$$r_2 = (m_1 \dot{l}_{10} + (m_2 + m_3) \cdot l_1)/m_{tot},$$
$$r_3 = (m_2 \cdot l_{20} + m_3 \cdot l_2)/m_{tot},$$
$$r_4 = m_3 \cdot l_{30}/m_{tot},$$
$$b = m_{tot}/(m_1 + m_2 + m_3),$$

$$(9)$$

where $l_1, l_2$ are the length of legs and thighs, $m_0, m_1, m_2, m_3$ are the masses of feet, legs, thighs, and trunk (HAT), ($m_{tot} = m_1 + m_2 + m_3 + m_4$), $x_0$ is the center of mass of the feet, $x_1, z_1$ are the coordinates of the ankle, and $l_{10}, l_{20}, l_{30}$ are the distances from the center of mass to the distal joints for the leg, thigh, and proximal joint for HAT. Coefficient $b$ accounts for the difference (the feet do not move during the dynamical experiments) in sensing the *COG* statically and dynamically. From this model, the six parameters of (9) are identified recursively with a non-linear least squares technique such as Levenberg–Marquardt, where *CÖG* is obtained approximately from numerical differentiation of *COG*. The actual small-scale leg was first identified, with results (statical and dynamical) contained in Figures 9 and 10.

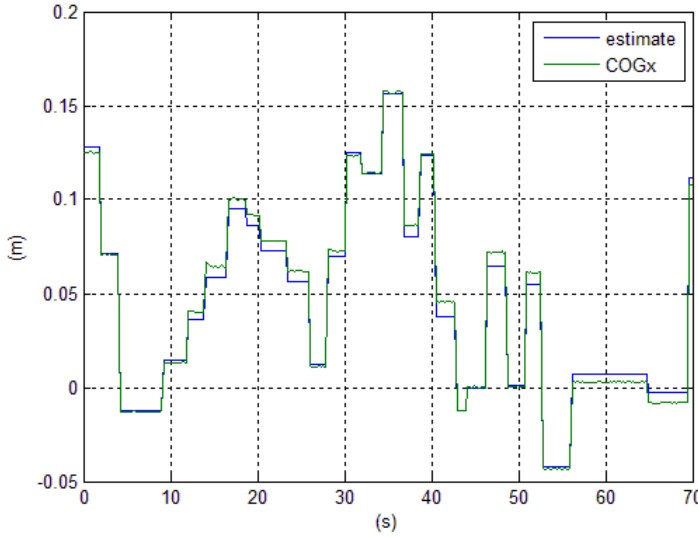

**Figure 9.** Results of estimating the statically equivalent serial chain (SESC) model in static experiments.

Then a control exercise was carried out, maintaining the *CoP* position fixed and the posture erect while performing an up-and-down motion (such as sit-to-stand) of the body. The results (*CoP$_x$* and *COG$_x$*), based on the identified model and the proposed control scheme, when a disturbing force is applied in the sagittal plane are shown in Figure 11. The action of the feedback on the *COG* to compensate the disturbance is clear.

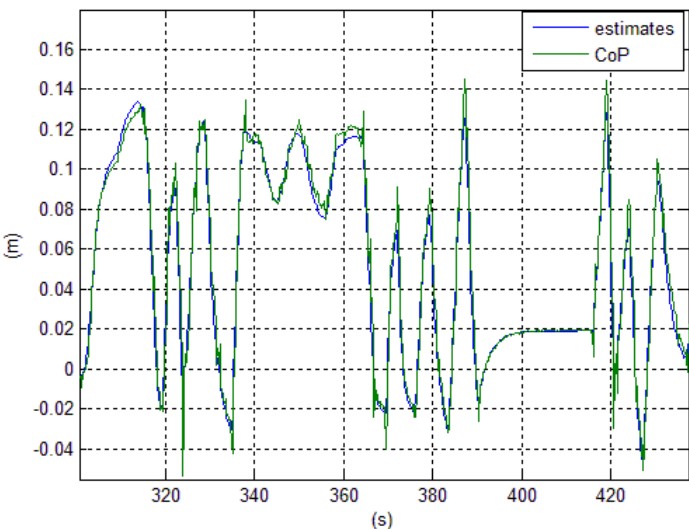

**Figure 10.** Results of estimating the SESC in dynamic experiments.

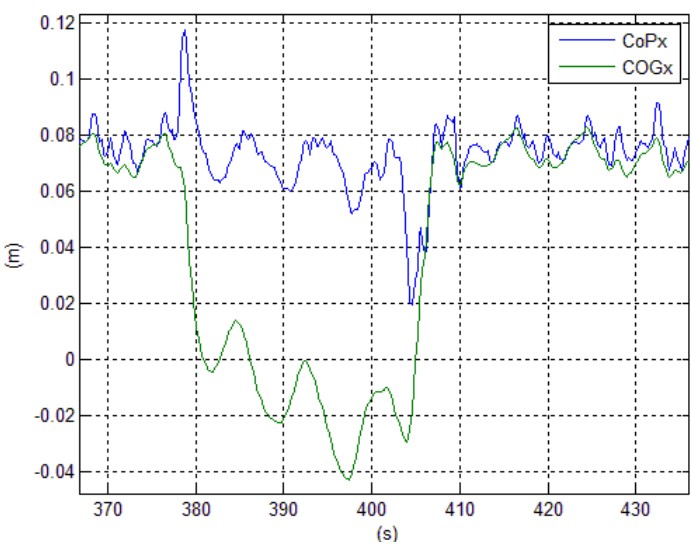

**Figure 11.** Maintaining the *CoP* position during a stand-to-sit like exercise in the presence of a disturbance force.

### 4.2. Simulation of a Stand-to-Sit Exercise

In order to validate the results on the linearized inverted pendulum of Section 3.3, the proposed control (with the same estimator and feedback parameters) was applied to the non-linear simulation of a multi-chain with 3 DOF, having as average the same $COG_z$. It represents a biped in the sagittal plane emulating the first phase of a stand-to-sit exercise to test the balance control of a future full-scale exoskeleton with a patient. While the pelvis is lowered from a standing posture to reach the chair and the trunk attitude assumes a natural bending forward, the $COG_x$ is shifted from heels to tips and a disturbance force is applied, as in the previous experiment of Figures 5 and 6. The animation of the exercise can be seen in Figure 12. The resulting response of the $COG - ZMP$ in Figure 13 is very similar to that of the linearized inverted pendulum. Particularly, in the final phase of the exercise, the reaction of the exoskeleton to preserve equilibrium against the push forward of the disturbing force is particularly visible.

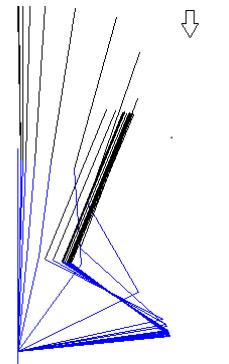

**Figure 12.** Animation of the 3 DOF kinematics during a stand-to-sit exercise.

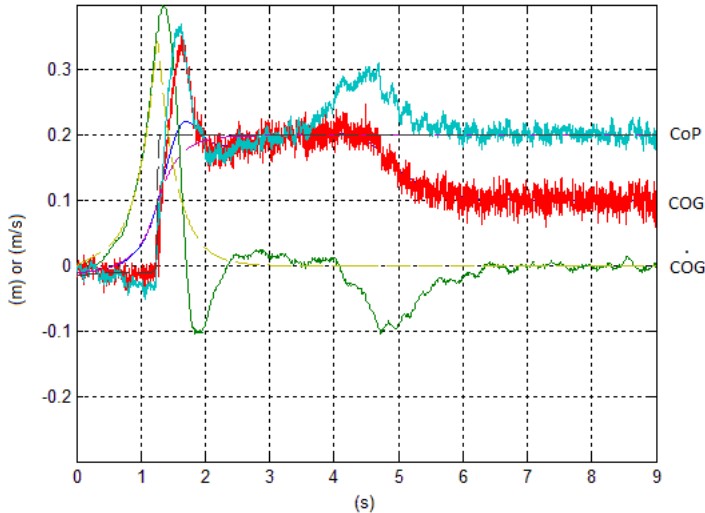

**Figure 13.** COG–ZMP during stand-to-sit and shift of *CoP* in a nonlinear 3 DOF kinematics-robust control.

Vice versa, Choi's original feedback with identical *COG* gain, applied to the nonlinear simulator, has not been able to guarantee stability.

## 5. Conclusions

In this paper, Choi's feedback for postural control of a biped robot, based on a linearized inverted pendulum model, has been revised. In practical situations, this feedback generates very undamped closed-loop dynamics. Then, the design problem was reformulated using state estimation and state feedback control. In fact, closing the loop with a state observer of the *COG* and exploiting velocity along with position and acceleration estimates guarantees a greater damping of closed-loop poles, with identical steady-state gain. However, to be effective in fusing *CoP* and *COM* measures, this observer needs to be extended to also estimate external disturbance forces, and the kinematic model of the *COM* needs to be tuned to the actual mass distribution. The former problem was tackled by a robust estimator based on an extended system embedding into the model unknown force disturbances, the latter by identifying a priori the SESC model of the mapping between joint angles and the *COM*. This a priori identification can also be repeated to maintain the mapping up to date in cases of changes in the weight distribution of the biped.

Two approaches to design the feedback were pursued: one is numerical, computing the state feedback for a given observer with a Levenberg–Marquardt algorithm. The second exploits integrally estimator and feedback obtained from a robust control regulator and adapts it to the tracking of

the preview signals. The results show similar performances, with good disturbance compensation. It must be emphasized that the adoption of an extended system with its weighting functions offers a formal technique to set the observer and feedback characteristics, guaranteeing the desired loop gain and bandwidth.

Robustness was shown by applying the control designed for a linearized inverted pendulum to two non-linear systems: a three-DOF kinematic chain of an actual mechanical small-scale leg and the simulation of an exoskeleton constraining a patient to perform a joint-legged stand-to-sit exercise in the sagittal plane. The proposed control correctly integrates the *COG* information, which is poorly reconstructed from joint measures and kinematics of the chain, with actual *CoP* measures, accommodating uncertainties in the model and unknown external force disturbances. Moreover, an identification procedure of the SESC model was proposed and tested.

COG–ZMP and linearized inverted pendulum models continue to be at the basis of balance control of bipeds. However, extended systems and SESC models have not yet been proposed in order to offer robustness to the approach.

The proposed $COG - ZMP$ control was successfully used by the authors for the balance of turning during walking of biped robots and for a more detailed and complete sit-to-stand exercise described in the companion paper [8]. In particular, future developments will consider haptic exoskeletons, where the action of the patient on some joints, through electromiographical signals, controls the motion of part of the degrees of freedom, while the automatic control discussed here guarantees balance acting on the ankles or on the hips.

Computing robust estimation and robust control, as well as the block diagrams present in the paper, were made with the design environment G++ developed by the authors described in [18] and that can be downloaded from [19]. However, the used technique is fairly standard in the robust control field and can be found in classical textbooks such as [14].

**Author Contributions:** Methodology, G.M. and M.G.; writing—original draft, G.M.; writing—review & editing, M.G.

**Funding:** This research has been partially supported by MIUR, the Italian Ministry of Instruction, University and Research through project ESOPO, and the Piedmont Region through project ESROB.

**Conflicts of Interest:** The authors declare no conflict of interest.

## Appendix A. Choi's Feedback Limitations

This appendix is devoted to showing that, in spite of stability, Choi's original feedback has a poor damping of the closed-loop dynamics in practical operating conditions. Applying the control strategy of Equation (4) to the model of Figure 1, we obtain a closed-loop system with three design gains $k_v, k_c, k_p$ and one coefficient $w_n$, as depicted in Figure A1. In classical linear control theory, it is customary to introduce disturbances in the input and output of the system and to study the closed-loop performance, analyzing the open-loop transfer function (t.f.), and the related closed-loop sensitivity functions linking the output to the input and output noises. The block diagram presents two partial feedback loops, on *COG* and on *ZMP*, that can be analyzed separately by opening (indicated with an $X$ in the block diagram) the two feedbacks one at a time and considering the other part of the system.

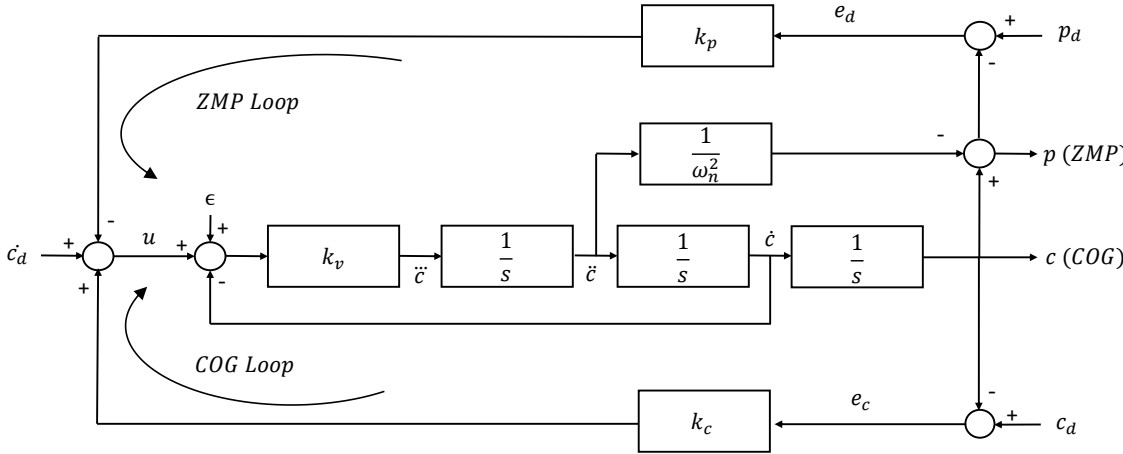

**Figure A1.** The COG–ZMP model of the linearized inverted pendulum with Choi's feedback control.

The functions related to the $ZMP$ loop are not particularly interesting: the feedback of the $ZMP$, although important for stability reasons, does not play any role in disturbance rejection and tracking error because its steady-state loop gain is always lower than 1. In fact, it is given by

$$G_{a_{zmp}}(s) = \frac{k_v k_p (1 - s^2/w_n^2)}{s^3 + k_v s + k_v k_c},$$
(A1)

where $k_p < k_c$ always holds.

Vice versa, the following t.f.s of the $COG$ loop are noteworthy: the open-loop transfer function $G_a(s)$, the output tracking error sensitivity $S(s)$, and the output sensitivity to input disturbances $G_{eq_{\epsilon-COG}}(s)$.

$$G_a(s) = \frac{k_v k_c}{s^3 + k_v k_p/w_n^2 s^2 + k_v s - k_v k_p}$$
(A2)

$$S(s) = \frac{s^3 + k_v k_p/w_n^2 s^2 + k_v s - k_v k_p}{s^3 + k_v k_p/w_n^2 s^2 + k_v s + k_v(k_c - k_p)}$$
(A3)

$$G_{eq_{\epsilon-COG}}(s) = \frac{k_v}{s^3 + k_v k_p/w_n^2 s^2 + k_v s + k_v(k_c - k_p)}$$
(A4)

In order to guarantee stability (negative real part of the roots of the third-order, closed-loop, characteristic polynomial appearing as denominator in Equations (A3) and (A4)), the following condition on the parameter $k_p$ must be satisfied:

$$\frac{w_n^2}{w_n^2 + k_v} k_c < k_p < k_c ,$$
(A5)

Note that condition (A5) is slightly different from Choi's result.

In order to have more insights about the closed-loop poles of Equations (A3) and (A4), consider the root locus, function of $k_v$, of the following open-loop transfer function:

$$G_{a_{k_v}}(s) = \frac{k_v(k_p/w_n^2 s^2 + s + k_c - k_p)}{s^3}.$$
(A6)

Note that the numerator of $1 + G_{a_{k_v}}(s)$ is exactly the characteric polynomial of (A3) and (A4). When $k_v \to \infty$, one real closed-loop pole $\to -\infty$, but the dominant closed-loop poles are complex conjugate and approach asymptotically the zeroes of the t.f. (A6), i.e., the root of the polynomial

$$s^2 + w_n^2/k_p s + w_n^2(k_c - k_p)/k_p, \tag{A7}$$

having the damping ratio

$$\zeta = \frac{w_n}{2}\sqrt{\frac{1}{k_p(k_c - k_p)}}. \tag{A8}$$

Moreover, the root locus shows that for any value of $k_v < \infty$, the damping ratio of the pair of dominant complex conjugate poles is always lower than that of these zeroes. From previous results, the following observations can be made:

- with high values of the $k_v$ gain, the dominant closed-loop poles depend on the pair $k_c, k_p$ only, as they are highly insensitive to $k_v$;
- the steady-state *COG* loop gain (A2) is proportional to the rate $k_c/k_p$ (independent from $k_v$);
- the steady-state gain of both sensitivities related to *COG* are inversely proportional to $k_c - k_p$;
- however, if the gain $k_c$, or more precisely, the difference $k_c - k_p$, increases, then the damping ratio of the dominant closed-loop poles decreases.

In conclusion, if a sufficiently high loop gain in the $COG - ZMP$ control system is imposed, with the feedback proposed by Choi, even if the closed loop remains stable, its behavior becomes highly undamped. However, a high loop gain, and hence a high value of $k_c$ is needed when a robust control has to be used in exoskeletons to improve postural equilibrium for ill or elderly people, in order to cope with uncertainties.

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
