# Peer review of "Estimation and Closed-Loop Control of COG/ZMP in Biped Devices Blending CoP Measures and Kinematic Information"

_robotics, doi:10.3390/robotics8040089_

Round 1
Reviewer 1 Report
The paper deals with Estimation and Closed Loop Control of COG/ZMP in
Biped Devices Blending CoP Measures and Kinematic Information. Paper supports research in area of bipped robotics.
The aim of this paper is to offer an engineering approach for the practical implementation of Choi’s method.
Technically it sounds very good. Graphical presentation of results is perfect.
Conclusion is very briefly. There is no mention about next future plans and trends in this area.
I have no other comments.
I recommend to accept it for publishing.
Reviewer 2 Report
This is an interesting paper on an interesting topic. That being said, there are a number of changes that must be made before this paper can be further considered for publication.
In general, the point of the paper seems to be lost amid lots of details about prior work. The abstract is too long and doesn't concisely explain the point of the paper. While it does touch on what the point of the paper is, it doesn't concisely explain what the specific contribution of the paper is.
The introduction, similarly, provides a lot of information about what has come before, but doesn't really explain why what is being presented in this paper is important. The paragraph beginning on line 59 has some information that may be relevant to this; however, it doesn't make this point clear. I would recommend significantly changing the introduction to explain the problem that you are attempting to address and the contribution that your work makes. Your next section focuses on prior work as well, and some of the content about prior work that is in the introduction could be moved into section 2 to make the introduction more clear.
The last paragraph of the introduction is very helpful, as it provides an overview of the rest of the paper. It should be made more concise.
In terms of the body of the paper, there are a few issues that need to be addressed throughout the paper. First, all of the flow chart pictures (Figs 1-4) are illegibly small and should be increased in size to be readable. Second, when equations are presented, whatever important details you would like the reader to gain from them should be said. Readers should not need to stop and analyze equations to follow your paper. Most won't. In this regard, the paper reads a bit too much like a text book. I would recommend looking at trying to reduce the number of equations that are presented to those that are essential and perhaps move some to an appendix, if they are supplemental to the point that you are trying to make. Third, please make sure that all figures are referenced in the text. Figure 14, in particular, doesn't seem to be mentioned anywhere.
I am also concerned about references 15 and 16. I would suggest that these may need to be added as supplemental content with this article. The drop box link may not be available in the future, and this would limit reader access to these resources which appear to be integral to this article. Alternately, you could upload these to another repository and provide a link to this.
A final area of consideration is to make your contributions more clear from the other work that you summarize throughout the paper. Sections 2 to 5 should be revised to make it clear as to what is new and what is pre-existing. Section 6 (conclusions) should make the specific contributions of this paper explicitly clear. While a discussion of future work is appropriate in this section, in addition to the discussion of what has been shown in this paper, other topics beyond this should be relocated to another section.
Largely section 2 to 5 seem strong; however, it will be easier to review these more fully once the changes described above are made and the authors' contributions are more clear.
This should make an interesting paper that will likely be very well received. I Would be happy to review a revision, if requested.
Reviewer 3 Report
This paper presents an engineering approach for the practical implementation of Choi’s method. It offers a more detailed understanding of the closed loop dynamics generated by the Choi’s feedback, building a model of the linearized inverted pendulum for an extended estimation, not only of COG and ZMP, but also of external disturbances, and for closed loop robust control design.
The scientific topic may be of interest to the readers. The paper is somehow well organized. Unfortunately, literature analysis is not enough and the contribution of the paper needs to be better introduced by pointing out the application of interest. In particular an introduction in the method section about the “ingredients of control theory” used to address bipedal robots instead of exoskeletons assisting walking task, would be beneficial to the reader. If the difference in algorithm from a high point of view may be clear since the begin of the paper, the reader would better follows the dissertation.
Finally, a specific experimental evaluation would be very beneficial.
Follows the Reviewer comments:
The abstract looks too long, many literature citations are present also, and the abstract should not be the summary of the introduction. The reviewer suggest shortening the abstract pointing out clearly the motivation and the paper contribution.
The introduction presents the scientific background, motivating the Authors to further extending the theory based on COG and ZMP. Have different methods been used in literature to address the same problem? One or two paragraph in the introduction may be used to give a full picture to the reader about actual methods and why the one proposed is the best approach. This will motivate the used of the proposed one.
Introduction misses possible robotic applications that will beneficial of this method.
The last introduction paragraph is too long. The methodologic information related to the algorithm used should be moved before the last paragraph. The last paragraph should be a summary of the paper contents.
Section 2 may improve readability if a diagram showing the variables described is added.
Line 95: it is missing the sentence subject.
Line 96: what do you mean with this sentence “The sagittal plane is considered in the following”
Line 98: “where p is the coordinate of the ZMP, c the projection of the COG on the ground, cz the height of the COG, g the acceleration of gravity. wn is the only parameter of the model of the simplified biped walking system”. c, cz and g sentences miss the verb is. The last sentence is not clear what the Authors want to imply.
Line 106: please add the number of the citation.
Line 112: “The third order model is needed to guarantee a realistic strictly proper system (position and acceleration are both present in the output) and representing, on the main time, with the internal speed loop the servo dynamics” sentence is not clear.
Line 116: kv is not present in eq 4. Please add a further equation showing the not obvious step implied.
Line 119: not clear what the authors mean with following “that can be analyzed separately, through several closed loop”.
Line 130: eliminate “the”
Line 152: what is the gain implied the following sentence: “A higher gain is needed when a robust control is to be used in exoskeletons to improve postural equilibrium for ill or elderly people”
Line 155: “The previous model does not take into account” is the one in fig 1? Please add reference.
Line 157: for the first time a possible application of the method is cited here “External forces are introduced when crutches are used or when the patient is sitting on a chair in a sit-to-stand exercise”. This theme should be presented first time in introduction and then recalled here.
Line 160: what the Authors add from the choy model in figure 3 related with the following sentence “extended system of fig. 3”. Please underline in the figure 3 the Authors contribution.
Line 166: are the Authors sure the term “noise €” in the paper is the standard letter used for control error
How are Section 4.1 and section 4.2 related to the previous section and the control diagram in figure 4?
Add legend on each figure (5,6,7,8) and for better reading alternate continuous line and dashed line. In addition, trend of different units should be plotted in separated charts or using two y-axes Matlab function.
The meaning of Figure 8 is not clear. Is this figure beneficial to the paper?
Line 228: there is some confusion on the methods and applications. The Authors jump from an application to another and it not clear enough what is the link, the theory bases and the difference between each approach. The reviewer suggest to make a step back and introduce, from a high point of view, the theory, the possible applications and the ingredients needed to address bipedal robot or exoskeletons (introduction and section 2 may be the right place). Then each following section may address the specific topic or application. From the reviewer looks two separate topic forced in a unique paper. Please clarify the issue.
Round 2
Reviewer 2 Report
This remains a very interesting paper and it is clear that the authors have made significant changes to it since the prior version. A number of additional areas need work before the paper can be published.
First, the paper still has considerable English usage issues. I would recommend finding a more fluent English speaker / writer to help with this. I think working with someone that you have direct access to interact with will be more helpful than trying to go through iterations with a copy editor trying to correct English language usage. At present, there are a number of English issues that could impact meaning and inadvertent changes to meaning can be difficult to catch as the copy editor won't have an knowledge of the field.
Second, I would suggest explaining the ZMP and inverted pendulum concepts very early in the paper as, at present, a reader without this background would likely be lost almost immediately. It would also be good to explain the implications of your work and what this means to someone working in robotics, more generally (as Robotics has a wide readership). This might also help people to find your article based on a broader set of keywords related to application of the work.
Third, as general housekeeping notes, I would recommend against making statements like "undeniable theoretical interest" (abstract) without explanation or citation. Also, there are a lot of things capitalized that don't need to be for example for "Zero Moment Point (ZMP)" only the Z needs to be capitalized, because it is at the beginning of a sentence (this is also in the abstract).
Forth, on lines 59-60, you state "the same approach has been used by the authors for the 60 balance of biped robots." You should make it clear as to where this is referring to.
Fifth, I would suggest making the introduction more focused on explaining what your particular work is. This should be able to be stated in one to three sentences. I would also suggest placing this right before the paragraph where you summarize the rest of the paper, as this is where readers will typically look to see what your paper's contribution is.
Sixth, I would remove the bullet points from the introduction and re-write this in paragraph form.
Seventh, I would suggest moving all of the background discussion not absolutely necessary to make your paper's contribution clear out of the introduction (which will help make the introduction easier to follow) into a background section. The Choi work discussion can become a subsection of this background discussion.
Eighth, Figures 4 to 8 should all be made larger to increase legibility. For Figures 4 to 7, there seems to be lots of extra white space around the figures that could be used so that they don't take up more page space.
Finally, and more generally, I would try to make this paper more generally accessible. Right now it seems like it would only be read or used by people working in a very specific area; however, you are dealing with a problem that has significant application to a lot of challenges in robotics and human assistance technology. If you assume a bit less prior knowledge on the part of readers and explain what you have shown and why this is important, I think you will find that you have significantly higher readership and citations for this paper.
Reviewer 3 Report
This paper has been revised extensively. The approach shows a practical implementation of Choi’s method of the closed loop balance control, fusing CoP and joint angle measures.
The paper is well organized and the presented background related with the control theory for balance control for bipedal robots is strong enough. However, this is not the same for the exoskeleton assisting walking task or sit-to-stand task. In fact, section 4.2 presents a simulation where the same control has been applied on a full leg exoskeleton assisting sit to stand task. This simulation does not consider the human factor (weight, internal forces and so on). The person is not evaluated into the simulation, thus, the reviewer considers not enough to state that the control approach is validated also for assistive exoskeletons.
The reviewer suggest to revise (or eliminate) the dissertation on the exoskeleton topic, in section 4.2 leaving it only in conclusions as future work.
